# DanSumT5: Automatic Abstractive Summarization for Danish

**Sara Kolding**
School of Communication and Culture
Aarhus University
Jens Chr. Skous Vej 2, 8000 Aarhus C
sarakolding@live.dk

**Katrine Nymann**
School of Communication and Culture
Aarhus University
Jens Chr. Skous Vej 2, 8000 Aarhus C
katrinesofiemn@hotmail.dk

**Ida Bang Hansen**
School of Communication and Culture
Aarhus University
Jens Chr. Skous Vej 2, 8000 Aarhus C
idabanghansen@gmail.com

**Kenneth C. Enevoldsen**
Center for Humanities Computing
Aarhus University
Jens Chr. Skous Vej 4, 8000 Aarhus C
kenneth.enevoldsen@cas.au.dk

**Ross Deans Kristensen-McLachlan**
Center for Humanities Computing
Aarhus University
Jens Chr. Skous Vej 4, 8000 Aarhus C
rdkm@cas.au.dk

## Abstract

Automatic abstractive text summarization is a challenging task in the field of natural language processing. This paper presents a model for domain-specific summarization for Danish news articles. DanSumT5 is an mT5 model fine-tuned on a cleaned subset of the DaNewsroom dataset comprising abstractive article-summary pairs. The resulting state-of-the-art model is evaluated both quantitatively and qualitatively, using ROUGE and BERTScore metrics, along with human rankings of the summaries. We find that although model refinements increase quantitative and qualitative performance, the model is still prone to factual errors. We discuss the limitations of current evaluation methods for automatic abstractive summarization and underline the need for improved metrics and transparency within the field. We suggest that future work should employ techniques for detecting and reducing errors in model output and methods for reference-less evaluation of summaries.

## 1 Introduction

### 1.1 Automatic text summarization

Automatic text summarization is the automatic generation of short text which condenses the most salient points of a longer text. Much of the research in this field to date has focused on automatic extractive summarization (El-Kassas et al., 2021), which directly extracts and concatenates sentences from the original text. Various methods have been developed for selecting sentences for extractive summaries. Some, such as TextRank (Mihalcea and Tarau, 2004), rely on measures of sentence importance; others rely on simple heuristics, such as sentence location. For instance, the simple LEAD-3 heuristic selects the first three sentences of a given article (Varab and Schluter, 2020). Extractive summarization thereby ensures grammaticality but tends to suffer from a lack of coherence. Additionally, extractive summaries can be afflicted with dangling anaphoras, in which an extracted sentence refers to a preceding sentence not included in the summary (Gupta and Gupta, 2019).

In recent years, the move towards deep learning in natural language processing (NLP) has brought to the forefront automatic abstractive summarization. Abstractive summaries paraphrase and condense the main points from the original text. This approach views summarization as a text-to-text problem on which models can be trained and fine-tuned. In many cases, abstractive methods enable more informative summarization, since rephrasing and compressing the text allows for more information to be conveyed by fewer sentences. However, factual credibility is not ensured, and the gener-

ated summaries may contain statements that are inconsistent with the original text (Maynez et al., 2020; Zhao et al., 2020).

The majority of existing work on both types of automatic summarization has been in English (Azmi and Al-Thanyyan, 2012; Khan et al., 2019). In this paper, we develop an abstractive text summarization model for Danish news data. We achieve state-of-the-art results by fine-tuning mT5 on a cleaned subset of the DaNewsroom dataset (Varab and Schluter, 2020) consisting of news articles and their corresponding abstractive summaries.

## 1.2 Previous work

This work builds on the authors' earlier attempts to develop an automatic summarization model for Danish, here referred to as $DanSumT5_{pilot}$. This previous work utilized the DaNewsroom dataset, but with smaller and less thoroughly filtered subsets, and without systematic hyperparameter search. An mT5 model trained on a subset representative of the full dataset performed similarly to extractive baselines validated on the full dataset (Varab and Schluter, 2020). However, upon further inspection, the resulting summaries were predominantly extractive, likely due to the amount of extractive summaries in the dataset. mT5 models trained on more abstractive subsets of the full dataset displayed more qualitatively and quantitatively abstractive behaviour, though the resulting summaries yielded lower quantitative results. The models predominantly generated short and repetitious summaries, possibly resulting in artificially inflated ROUGE performance.

The previous studies yielded two tentative but significant insights. Firstly, we established that fine-tuning an mT5 model capable of producing abstractive summaries does not ensure abstractive summarization. Secondly, our work emphasized the need for employing more nuanced quantitative metrics, as well as qualitative inspection of model output, to determine the quality and abstractiveness of the generated summaries.

The current work expands on these previous efforts by implementing several changes to the data, model, evaluation, and fine-tuning procedure.

### 1.2.1 Multilingual language models

In this work, we use mT5 (Xue et al., 2021), a multilingual T5 architecture (Raffel et al., 2020) pre-trained on data from 101 languages. There are several reasons for this, beyond the T5 architecture being well-suited to text summarization tasks. While there is evidence to suggest that monolingual models generally perform better on monolingual tasks (Nozza et al., 2020; Popa and Stefănescu, 2020; Rust et al., 2021) (see also section 6), multilingual models seem to increase performance for smaller languages, likely by leveraging cross-lingual transfer (Conneau et al., 2020; Lauscher et al., 2020). Additionally, it has been suggested that this effect depends on the size of the language-specific vocabulary during pre-training, as well as lexical and typological proximity between included languages (Lauscher et al., 2020; Rust et al., 2021). Indeed, target language performance appears to be related to both size of the target-specific pre-training corpora, as well as linguistic similarity between the target and source language (Arivazhagan et al., 2019; Lauscher et al., 2020). We contrast the multilingual architecture with the recently developed monolingual model, DanT5 (Ciosici and Derczynski, 2022).

## 1.3 Quantitative evaluation metrics

ROUGE-1 (R-1), ROUGE-2 (R-2) and ROUGE-L (R-L) denote the co-occurrence of unigrams (R-1) and bigrams (R-2) in the generated and reference summary (Lin, 2004; Varab and Schluter, 2020), as well as the Longest Common Subsequence (R-L) (Briggs, 2021; Lin, 2004). R-L is thereby the only ROUGE measure capable of considering syntax, since it rewards longer identical sequence overlaps. R-1 and R-2 might inflate performance, even if the generated summary is syntactically incoherent, if numerous co-occurrences from the reference summary are present. ROUGE scores have displayed some correlation with human evaluation of summary fluency and adequacy (Lin and Och, 2004), and are the most commonly used metric for automatic summarization.

It should be noted that ROUGE scores assess lexical overlap of strings, with no reference to semantic similarity. A qualitatively acceptable abstractive summary could consist of completely novel strings, with no overlaps relative to the source text, which would be penalized by the ROUGE metric. Alternative evaluation metrics instead utilize the semantic and syntactic relations captured in contextualised word embeddings produced by transformer-based architectures, such as

BERT (Devlin et al., 2019). BERTScore (Zhang et al., 2020) calculates the similarity between generated and reference summaries as the sum of cosine similarities between the contextual embeddings of tokens. Greedy matching is used to compute a score for each embedding in the generated summary and the most similar embedding in the reference summary. In what follows, we evaluate model performance using a combination of both ROUGE and BERTScore.

## 2 Dataset

### 2.1 DaNewsroom dataset

The DaNewsroom dataset (Varab and Schluter, 2020), inspired by the English Newsroom dataset (Grusky et al., 2018), is currently the only publicly available dataset for Danish summarization. The dataset consists of 1.1 million article-summary pairs published over the past 20 years in various Danish news outlets. The summaries were retrieved using a metadata tag and, in most cases, correspond to the subheading of the article.

#### 2.1.1 Data cleaning

To quantify the degree of abstractiveness of a summary, we use the density score (Varab and Schluter, 2020; Grusky et al., 2018), where lower scores indicate more abstractive summaries, and higher scores indicate increasingly more extractive summaries, i.e. summaries containing longer identical sequence overlaps with the original text. Density is defined as:

$$Density(A, S) = \frac{1}{|S|} \sum_{f \in F(A,S)} |f|^2$$

Where *(A,S)* is an instance pair of an article and a summary, and *F(A,S)* is the set of extractive fragments *f* of longest common sequences of tokens in *A* and *S*.

Based on this density measure, the reference summaries in the DaNewsroom dataset are primarily extractive. For the purpose of abstractive summarization, we follow binned density threshold categories (Grusky et al., 2018; Varab and Schluter, 2020), and filter our dataset to contain only abstractive text-summary pairs with a density score between 0 and 1.5.

Several reference summaries in DaNewsroom are short and/or incomplete, as in multiple cases, the web scraping used to collect the dataset extracted incorrect or partial reference summaries.

Examples of short or single-word summaries include '2008', 'Et', 'Behandling', and 'P'. Similarly, the dataset also contains some extremely short and/or incomplete articles. In particular, a number of the articles are just one-liners about television scheduling or a paywall:

- Der er lukket for nye kommentarer til denne artikel (*This article is closed for new comments*)

- Svanerne i Slotsparken har fået fem unger (*The swans in Slotsparken have had five cygnets.*)

- 'DR1 — Tirsdag d. 26. august kl. 20:00 - 20:45'. (*DR1 — Tuesday 26 August at 20:00 - 20:45*)

On the other hand, some of the articles are very long. In some cases, the web scraping concatenated the article with a long thread of comments, resulting in articles consisting of several thousand tokens, including English content. We further cleaned the dataset by filtering the summaries and articles based on token length, with lower and upper cutoffs defined respectively as the 2nd and 98th percentiles. Additionally, the data was standardized using ftfy (Speer, 2019) and filtered using various heuristic quality filters, including filters for removing repetitious text, text with a high ratio of non-alphabetic tokens, and text with less than two stop words. This filtering was performed using an implementation similar to textdescriptives (Hansen and Enevoldsen, 2023) and follows an approach similar to Rae et al. (2022). The resulting cleaned subset contained 258,146 abstractive article-summary pairs. Figure 1 shows the distribution of article and summary token lengths before and after this filtering procedure.

## 3 Model

### 3.1 Infrastructure

We fine-tuned all of our models using the Transformers library (Wolf et al., 2020) and PyTorch (Paszke et al., 2019) as the back end on 4 RTX8000 GPUs. We used Weights and Biases (Biewald, 2020) for experiment tracking and visualizations.

### 3.2 Training and models specifications

For our hyperparameter tuning, we trained for 1800 steps with a batch size of 120, and validated

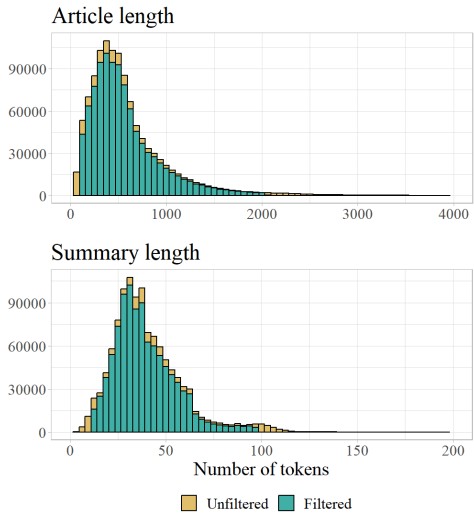

Figure 1: Distribution of summary and article token lengths before and after data cleaning.

with the same batch size, using the full validation set. All models were trained on our filtered dataset using an 80-10-10 split. Due to computing constraints, the hyperparameter search was performed using small-sized models ( 300M parameters) and for a limited number of steps. The hyperparameter search showed only a few consistent trends and based on this we chose a learning rate of $3.0 \times 10^{-4}$, a dropout rate of $0.01$, and a polynomial learning rate schedule. For information about the hyperparameter search, see Appendix A[1].

Articles and reference summaries were truncated to a maximum length of 1024 and 128 tokens, respectively. Mixed precision training was employed to lower the memory impact during training. The models were trained using the AdamW optimiser (Kingma and Ba, 2017) with a polynomial learning rate schedule using a learning rate of $3.0 \times 10^{-4}$, 2000 warmup steps, and a realized batch size of 16 (a batch size of 8 with an accumulation step of 2). Due to memory constraints, a smaller batch size was used for training as compared to hyperparameter tuning. The models were trained for ten epochs with a dropout rate of 0.01.

For decoding, we used beam search with two beams and a repetitive 3-gram penalty. Additionally, to encourage longer summaries, we set minimum and maximum generation lengths of 9 and 128 and employed a length penalty of 5. The best model was determined according to the cross

---

[1]Or see the Weight and Biases dashboard at https://tinyurl.com/3zfuf6vx

entropy loss, and the resulting model was tested using a held-out test set of abstractive reference summaries, according to binned density. From this sample of 25,830 article-summary pairs, generated summaries were evaluated by calculating mean density and mean F1-scores for R-1, R-2, R-L (Lin, 2004), and BERTScore (Zhang et al., 2020). The large XLM-RoBERTa model was used to create the embeddings for the BERTScore results (Conneau et al., 2020).

The mT5 model has been made available in different sizes, with larger sizes generally leading to improved performance (Xue et al., 2021). We fine-tuned three different mT5 model sizes; small, base and large, as well as a small DanT5 for comparison. All other parameters were kept constant.

Additionally, we validated the performance of the LEAD-3 and TextRank approaches on our test set. Importantly, both of these methods are extractive, and thus are not necessarily meaningful comparative baselines for abstractive summarization. Still, they provide a benchmark for quantitative comparison, though it should be noted that the test set comprises abstractive reference summaries. We also include performance metrics for DanSumT5$_{pilot}$, which was the best performing abstractive summarizer from our previous unpublished work.

## 4 Results

### 4.1 Quantitative results

Table 1 shows R-1, R-2, R-L, and BERTScore results for our three fine-tuned mT5 models and the DanT5 model. Additionally, we present a version of our previous work, DanSumT5$_{pilot}$, trained and evaluated on the same data splits of our current work. We compare these results to the performance of two extractive baseline models, LEAD-3 and TextRank, on the test set.

Mean F1 scores for all metrics are reported. Additionally, 95% confidence intervals for all metrics are calculated using bootstrap resampling with 1000 samples, following the original ROUGE Perl implementation (Li, 2020).

Table 2 shows the mean F1 density scores for the aforementioned models. 95% confidence intervals are calculated using bootstrap resampling with 1000 samples.

The best performing model according to quantitative metrics is DanSumT5$_{large}$. Furthermore, this model generates the lowest-density sum-

| Model | R-1 | R-2 | R-L | BERTScore |
|---|---|---|---|---|
| LEAD-3 | 18.31 [18.21, 18.42] | 4.60 [4.55, 4.66] | 12.31 [12.25, 12.39] | 86.77 [86.75, 86.79] |
| TextRank | 14.80 [14.71, 14.89] | 2.82 [2.78, 2.87] | 10.03 [9.98, 10.09] | 85.86 [85.84, 85.88] |
| DanT5$_{small}$ | 20.68 [20.54, 20.82] | 5.92 [5.83, 6.02] | 15.55 [15.44, 15.67] | 88.06 [88.04, 88.09] |
| DanSumT5$_{pilot}$ | 19.74 [19.57, 19.90] | 6.63 [6.52, 6.74] | 16.71 [16.57, 16.85] | 88.02 [87.99, 88.05] |
| DanSumT5$_{small}$ | 21.42 [21.26, 21.55] | 6.21 [6.11, 6.30] | 16.10 [15.98, 16.22] | 88.28 [88.26, 88.31] |
| DanSumT5$_{base}$ | 23.21 [23.06, 23.36] | 7.12 [7.00, 7.22] | 17.64 [17.50, 17.79] | 88.77 [88.74, 88.80] |
| DanSumT5$_{large}$ | **23.76 [23.60, 23.91]** | **7.46 [7.35, 7.59]** | **18.25 [18.12, 18.39]** | **88.97 [88.95, 89.00]** |

Table 1: Mean F1 ROUGE and BERTScore performance by model with 95% bootstrapped confidence intervals. Best score is highlighted in bold. Abstractive and extractive methods are delineated.

| Model | Density |
|---|---|
| LEAD-3 | 26.01 [25.84, 26.18] |
| TextRank | 32.23 [32.07, 32.40] |
| DanT5$_{small}$ | 2.91 [2.89, 2.93] |
| DanSumT5$_{pilot}$ | 2.76 [2.74, 2.78] |
| DanSumT5$_{small}$ | 2.66 [2.65, 2.68] |
| DanSumT5$_{base}$ | 2.32 [2.30, 2.34] |
| DanSumT5$_{large}$ | 1.91 [1.90, 1.93] |

Table 2: Mean F1 density of generated summaries for the different models with 95% bootstrapped confidence intervals. Summaries with a density below 1.5 are considered abstractive, while summaries with a density above 8.19 are considered extractive.

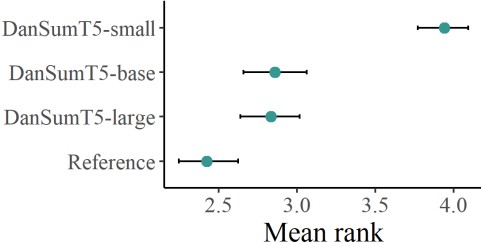

Figure 2: The mean rank obtained for each model through human evaluation. Error bars display the 95% bootstrapped confidence intervals.

maries, which indicates more abstractive summaries.

## 4.2 Human evaluation

To manually evaluate the models, we randomly sampled 100 articles in the test set, along with both reference and generated summaries. Two of the authors were selected as raters and tasked with reading the articles and ranking the four summaries (DanSumT5$_{small}$, DanSumT5$_{base}$, DanSumT5$_{large}$, and the reference summary) according to preference[2]. Both raters are women educated to master's level, and both are native speakers of Danish. The rating was blind, with ratings being unaware of the origins of the summaries. The raters were asked to rate based on preference. This rating procedure does not quantify the objective quality of each summary but instead evaluates the relative quality between summaries. Consequently, we cannot determine

why the summary was preferred, or which aspect of the summary contributed to the decision, such as grammaticality or factuality. The two raters had an agreement of 74.8% (95% CI: [71.2, 78.2]). The agreements are calculated based on each pair of comparisons, i.e. every possible comparative relation between summaries. Both raters generally preferred the reference summaries over the generated summaries, as shown in Figure 2.

The best performing model according to subjective evaluation was DanSumT5$_{large}$. The generated summaries are generally grammatically correct and cover the main content of the article, though they tend to suffer from factual inconsistencies (See Appendix B for five randomly chosen examples of reference and model-generated summaries).

## 5 Discussion

### 5.1 Evaluation

As seen in Table 1, DanSumT5$_{large}$ achieves higher ROUGE scores and BERTScores than our previous work DanSumT5$_{pilot}$. We have thus achieved state-of-the-art results for Danish abstractive summarization. Furthermore, DanSumT5$_{large}$ generates relatively low-density

[2]See GitHub for the full ratings. Located in the data folder https://github.com/Danish-summarisation/DanSum.git

summaries. Notably, the mean density falls just outside the abstractive binned density category. Still, the generated summaries are of lower density than those generated by the other models, or by the extractive comparisons. All DanSumT5 models outperform extractive baselines, likely at least partly due to the test set comprising only abstractive reference summaries.

Human evaluation reveals that DanSumT5$_{large}$ generally creates summaries that were highly rated relative to the other summaries. For instance, the DanSumT5$_{large}$ summaries were preferred over the reference summaries in 21.43% of cases. One limitation of summaries generated by all DanSumT5 models is that many of them suffer from factual inconsistencies, as illustrated in Appendix B, Table 7 where DanSumT5$_{small}$ states that artists were hit by a snowstorm, though in actuality, an artificial snowstorm was part of their performance.

## 5.2 Limitations and Future Directions

### 5.2.1 ROUGE and BERTScore

Given the inherent limitations of ROUGE scores for evaluating abstractive summaries, we opted to employ BERTScore which does not penalize lexical diversity. Since BERTScore is fully differentiable, it could be used to compute a loss metric for optimisation of both training and evaluation (Zhang et al., 2020). However, some limitations of ROUGE also apply to BERTScore, inasmuch as precision, recall and F1 scores depend on the length of the generated and reference summary. For both ROUGE and BERT metrics, generating very short summaries or using very short reference summaries might inflate performance, since co-occurrences in short sequences are disproportionately rewarded for high relative overlap between reference and model output. Additionally, a high BERTScore is also not a guarantee of factual or grammatical consistency.

Many quantitative evaluation metrics reward similarity with the reference summary; however, this might not be optimal. For instance, since many reference summaries in the DaNewsroom dataset correspond to isolated article subheadings with dangling anaphoras, these issues could transfer to generated summaries. A possible way to remedy this would be to utilize anaphoric information, for instance by checking co-references of the generated summary, and locating errors relating to anaphoric resolution (Steinberger et al., 2007; Sukthanker et al., 2020). Also, despite additional filtering, some reference summaries and articles in the dataset were still incomplete. Mismatched or incomplete summary-reference pairs complicate the task of the model, leading to nonsensical or unrelated outputs: In Appendix B, Table 6, the model-generated summaries contain factual errors and nonsensical phrases, while the reference summary appears unrelated to the accompanying article (full article not included due to copyright). This is because the article was incompletely sampled, whereby critical information was omitted. In these cases, comparisons between generated and reference summaries are illogical. Alternative quantitative metrics suggest omitting the reference summary and evaluating performance using only the original text, for instance by calculating the increase in task performance gained by access to the generated summary (Vasilyev et al., 2020).

Recent research shows that most quantitative metrics do not correlate well with human evaluations of generated summaries in important dimensions such as coherence, consistency, fluency, and relevance (Fabbri et al., 2021; Liu et al., 2017). Indeed, it has also been argued that there is no best practice for reliable human evaluation of summaries, and that human evaluations often do not correlate with other human evaluations of the same summaries (Fabbri et al., 2021; Iskender et al., 2021). Though human evaluations are often presented as the gold standard, evaluator demographics, expertise, and task design hugely affect human evaluations (Harman and Over, 2004; Louis and Nenkova, 2013). Different summaries may focus on different aspects of the same article, with no way to objectively conclude which is better. Consequently, future research might benefit from optimizing reference-free metrics to evaluate generated summaries independent of "gold-standard" counterparts.

### 5.2.2 Quantifying abstractiveness

There is no clear definition of what counts as an 'abstractive summarization model'. Many studies on abstractive summarization do not report or evaluate the density of the summaries in their dataset, or of their model-generated summaries. A high ROUGE score could correspond to a predominantly extractive or simply very short reference or generated summary, thereby inflating ROUGE performance. We note that our model-generated

summaries have a wide range of density scores, with the largest model producing more abstractive (low-density) summaries. Since abstractive summarization allows high lexical diversity, it is not likely to achieve as high ROUGE scores as extractive summarizers, and thereby low ROUGE performance need not be strongly indicative of poor abstractive summarization.

Finally, existing work tends not to present translated examples of model output. While translation might not be the optimal reflection of non-English summarization, it allows readers to evaluate an approximation of the qualitative results. Lack of transparency, therefore, makes it extremely challenging to evaluate whether the reported summarizers are truly abstractive, or whether high ROUGE performance reflects extractive, short or repetitive summaries.

### 5.3 Model limitations

#### 5.3.1 Factors limiting practical implementation

Automatic summarization requires the model to be factual to the source text, especially for real-world practical implementations. However, none of the evaluated metrics considers the factual correctness of a generated summary (Falke et al., 2019) and does not reward the model for being factually faithful (Maynez et al., 2020).

Since DanSumT5 sometimes generates summaries with obviously incorrect content (see Appendix B), it is unsuitable for practical implementations where factual accuracy is important, such as summaries of news articles. One possible solution could be to use a separate system to detect such errors in the generated summaries (Falke et al., 2019), such as already existing systems for detection of errors related to quantities (Zhao et al., 2020). Fine-tuning according to this metric, or even using it for reinforcement learning, could alleviate concerns around accuracy and quality. Another approach to enhancing factual accuracy uses question asking to evaluate factual consistency by checking if the generated summary and article yield the same answer (Wang et al., 2020). Future work could extend this approach to Danish.

#### 5.3.2 Data limitations and considerations

Many of the reference summaries suffer from dangling anaphoras since they are scraped from the article's subheading, often lacking the context of the

title. These were likely not intended to be read as summaries, or even read in isolation from the article's title. This underlines the importance of data quality, since it defines the upper limit for model performance. We found that most of the 100 DanSumT5 summaries inspected for evaluation avoid dangling anaphoras, likely due to only a minority of the dataset suffering from this artefact, and could thus be argued to be better than the reference summaries in this aspect. For example, in the Appendix B, Table 5 the reference summary refers to "the superstar", while the model summaries mention the actor by name.

The current paper demonstrates that it is possible to fine-tune a multilingual model to create a performant text summarization model for a specific domain of Danish language. Other directions for future work must therefore include further experimentation with different datasets. The practical costs of creating high-quality datasets for this task are a challenge for a language such as Danish which is relatively model-rich, compared to similarly sized languages, and data-poor in terms of high-quality data, compared to larger languages. However, as shown in this paper, good results can be obtained by fine-tuning a multilingual model on a web-scraped dataset with minimal data cleaning.

## 6 Negative Results

During the training of these models, we attempted a few additional ideas, most of which were shown to be unpromising. This section briefly describes these attempts:

1. As suggested by (Abdaoui et al., 2020) we reduced the model by restricting the vocabulary to Danish and English tokens. Meaningful tokens were estimated using a filtered version of the Danish Gigaword (Strømberg-Derczynski et al., 2021)[3] and English Gigaword (Graff and Cieri, 2003). Reducing the model size allowed us to train these models with a larger batch size. While the reduced size led to similar performance to the original architecture for the small and base-sized models, the large pruned model proved highly unstable during training.

2. During the early phases of development, we also experimented with the recently released

---

[3]More information about the specific dataset can be found at `https://huggingface.co/datasets/DDSC/dagw_reddit_filtered_v1.0.0`

Danish T5 model, DanT5 (Ciosici and Derczynski, 2022). As part of the initial grid search over hyperparameters, we discovered that this monolingual model performed consistently worse compared to the similarly sized multilingual mT5 models. Table 1 illustrates this disparity on the full dataset. This could be due to DanT5's novel warm-starting approach which utilises an English T5 model checkpoint, but further experimentation is required in this area.

3. This work seeks to train models for abstractive summarization and thus filters out extractive summaries from the training data based on a density threshold. We experimented with lowering the density threshold for the training set summaries to include increasingly more extractive summaries. While we obtained a lower loss when including moderately more extractive references, the resulting summaries were notably more extractive.

4. In an attempt to avoid potential overfitting on the filtered dataset, we also experimented with training a large mT5 model for only 1 epoch using similar hyperparameters with the exception of a dropout rate of 0.1. In the human evaluation, this model placed slightly below the base-sized model of DanSumT5.

# 7 Conclusion

This paper presents DanSumT5, a set of models achieving state-of-the-art results in automatic abstractive summarization for Danish news articles. These results were achieved by fine-tuning mT5 models and implementing more thorough cleaning of the DaNewsroom dataset. We present state-of-the-art ROUGE and BERTScore performance for Danish abstractive summarization with our DanSumT5$_{large}$. Human inspection of the relative quality of the generated summaries revealed that they were generally grammatical and coherent. We discuss several limitations of the quantitative metrics, emphasizing that ROUGE penalizes lexical diversity inherent to abstractive summarization, while high quantitative performance could obscure low qualitative performance. This emphasizes the need for more transparency in the field, and we argue that research should include more nuanced metrics, as well as manual evaluation of the density and overall quality of the

model output. Limitations of our work include data quality and the prevalence of factual errors in the generated summaries. All code related to this project is open-sourced via Github[4], and the model is made freely available via Huggingface for public use[5].

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

# Appendices

## A   Hyperparameters appendix

Table 3 shows the hyperparameter search space.
For sampling, we use the Tree-structured Parzen
Estimator implemented in Optuna. The search
showed only a few consistent trends, and based
on this, we chose a learning rate of $3.0 \times 10^{-4}$,
a dropout rate of 0.01, a polynomial learning rate,
and the mT5 model set.

For more information, see the Weights and
Biases dashboard at `https://tinyurl.com/3zfuf6vx`.

| Models | DanT5$_{small}$, mT5$_{small}$ |
|---|---|
| **Learning Rate** | 0-0.1 |
| **Learning Rate Schedule** | Polynomial, Linear, Cosine, Constant |
| **No Repeat N-Gram Size** | 3-5 |
| **Length Penalty** | 4-7 |
| **Max Grad Norm** | 1-10 |
| **Data Quality** | {True, False} |

Table 3: Hyperparameters for Model Search.

## B Examples of generated summaries appendix

This appendix consists of five randomly sampled examples of automated summaries generated by our models, along with the corresponding reference summary from the DaNewsroom dataset.

| Origin | Summary |
|---|---|
| **Reference** | Viborg-Esbjerg 0-5 Et hattrick fra den tidligere OB-spiller Rajko Lekic sender Esbjerg på førstepladsen i Superligaen. 
 *Viborg-Esbjerg 0-5 A hat-trick from former OB player Rajko Lekic sends Esbjerg to first place in the Super League.* |
| DanSumT5$_{large}$ | Esbjerg fører nu Superligaen efter to sejre på stribe og en 5-0-sejr over Viborg på egen bane. 
 *Esbjerg now leads the Super League after two wins in a row and a 5-0 win over Viborg at home.* |
| DanSumT5$_{base}$ | Esbjerg satte Viborg på plads med en 5-0 sejr over Esbjerg, der dog stadig har svært ved at score i de seneste to kampe i Superligaen. *Esbjerg put Viborg in their place with a 5-0 victory over Esbjerg, who still have difficulty scoring in the last two games in the Super League.* |
| DanSumT5$_{small}$ | Esbjerg satte Viborg på plads i Superligaen efter en skuffende 7.-plads. Viborg er ude af sæsonen med en 6-0-sejr på 6-0 over Viborg. - Sport *Esbjerg put Viborg in place in the Super League after a disappointing 7th place. Viborg is out of the season with a 6-0 win of 6-0 over Viborg. - Sports* |

Table 4

| Origin | Summary |
| --- | --- |
| **Reference** | Superstjernen havde sine hunde med til Australien på ulovlig vis – det kan komme til at koste dyrt
*The superstar brought his dogs to Australia illegally – it could be expensive* |
| DanSumT5$_{large}$ | Hollywood-stjernen risikerer fængselsstraf for at have smuglet sine to hunde til Australien, hvor han filmede med "Pirates of the Caribbean"-filmene. • Johnny Depp og Amber Heard sender hundene hjem • Billedserie: Johnny Deeps venner i Australien • Depp: Jeg elsker dem • Politiker: De er slemme at smide os ud • Politi: De elsker hundene • Politiet: De har svært ved at slippe for straf • Depps hunde er farlige for
*The Hollywood star faces jail time for smuggling his two dogs to Australia, where he filmed the "Pirates of the Caribbean" movies. • Johnny Depp and Amber Heard send the dogs home • Photo series: Johnny Depp's friends in Australia • Depp: I love them • Politician: They are mean to throw us out • Police: They love the dogs • Police: They have a hard time escaping punishment • Depp's dogs are dangerous to* |
| DanSumT5$_{base}$ | Johnny Depp og hans kone Amber Heards ulovlige medbringelse af deres hunde til Australien kan koste dem op til ti års fængsel. - Kendte
*Johnny Depp and his wife Amber Heard's illegal bringing of their dogs to Australia could cost them up to ten years in prison. - Famous people* |
| DanSumT5$_{small}$ | Johnny Depps ulovlige medbringning af deres to hunde i Australien må måske gå i fængsel for deres ulovlig medbringing.
- Film og TV
*Johnny Depp's illegal bringing of their two dogs in Australia may have to go to jail for their illegal bringing. - Movies and TV* |

Table 5

| Origin | Summary |
|---|---|
| **Reference** | Hvad får voksne mennesker over 40 til at bruge flere timer om ugen på at vække eventyret til live i den virtuelle verden? De gængse svar på spørgsmålet bunder i fordomme, mener to inkarnerede computerspillere. Her følger deres egen forklaring fra det virtuelle univers

*What makes adults over 40 spend several hours a week bringing the adventure to life in the virtual world? The usual answers to the question are rooted in prejudice, say two inveterate computer gamers. Here follows their own explanation from the virtual universe* |
| DanSumT5$_{large}$ | Interview: Det er svært at forestille sig, at livet uden for murene i Danmark kan blive bedre, end man umiddelbart skulle tro. Det kræver i hvert fald en særlig indsats, siger forfatteren Mathiador, der er aktuel med bogen "Kroniken" om en ung mands liv i skoven bagved ved Grønland. Han fortæller, hvordan det er at blive klogere på livet ude i verden, og hvorfor det er sådan en

*Interview: It is hard to imagine that life outside the walls of Denmark can be better than one would initially think. In any case, it requires a special effort, says the author Mathiador, who is current with the book "The Chronicle" about a young man's life in the forest behind in Greenland. He tells what it is like to learn more about life out in the world, and why it is such a* |
| DanSumT5$_{base}$ | Interview: Journalisten Mathiador er en af de danske forfattere, der har skrevet bogen "journalisten", der handler om livet som journalist. Han er aktuel med en ny bog om bogen, der er fyldt med højdepunkter, fortæller han i denne uges udgave af'Journalisten'. I dag er han aktuel i en bog, som er baseret på bogen 'Avisen', som handler direkte om journalisten og skuespilleren, der skriver klummer i Lokalavisen Assens.

*Interview: The journalist Mathiador is one of the Danish authors who wrote the book "the journalist", which is about life as a journalist. He is current with a new book about the book, which is full of highlights, he tells in this week's edition of 'Journalisten'. Today he is current in a book which is based on the book 'Avisen', which is directly about the journalist and actor who writes columns in the local newspaper Assens.* |
| DanSumT5$_{small}$ | Den danske forfatter og forfatter Mathiador fortæller om sin evne til at blive kendt for sin kærlighed til livet i Danmark. Han er kendt som " Journalisten", men i dag er han aktuel med at gøre det svært for ham at komme i gang med at finde en plads i rummet, hvor han er i centrum af en række af verdens mest prestigefyldte kunstnere, der elsker at skabe kærligheden. Det er en sjælden kærlighed, når han rejser sig til landet.

*The Danish writer and author Mathiador talks about his ability to become known for his love of life in Denmark. He is known as "The Journalist", but today he is current with making it difficult for him to start finding a place in the room where he is at the center of a number of the world's most prestigious artists who love to create love. It is a rare love when he stands up to the country.* |

Table 6

| Origin | Summary |
| --- | --- |
| **Reference** | De danske vikinger leverede en fejlfri præstation under den første prøve ved Eurovision. Men ikke alle elementer fungerede optimalt. |
| | *The Danish Vikings delivered a flawless performance during the first rehearsal at Eurovision. But not all elements worked optimally.* |
| DanSumT5$_{large}$ | Det danske Grand Prix-hold havde problemer med sneen til prøverne tirsdag eftermiddag. |
| | *The Danish Grand Prix team had problems with the snow for the rehearsals Tuesday afternoon.* |
| DanSumT5$_{base}$ | Rasmussen og hans vikinger havde tirsdag første prøve i Eurovision Song Contest, men sneen var det største problem. Se videoen her på siden og læs mere. |
| | *Rasmussen and his Vikings had their first rehearsal in the Eurovision Song Contest on Tuesday, but the snow was the biggest problem. Watch the video on this page and read more.* |
| DanSumT5$_{small}$ | Det danske Grand Prix-håb Rasmussen blev tirsdag eftermiddag ramt af en snestorm, der ifølge DRs underholdningschef blev en rigtig god oplevelse for de danske deltagere i Melodi grand Prix i Portugal. |
| | *The Danish Grand Prix hope Rasmussen was hit by a snowstorm on Tuesday afternoon, which according to DR's entertainment manager was a really good experience for the Danish participants in the Melodi Grand Prix in Portugal.* |

Table 7

| Origin | Summary |
| --- | --- |
| **Reference** | Flere anmeldere roser filmen om forhistorien til Harry Potter for at være underholdende og velsmurt. 
 *Several reviewers praise the film about the prequel to Harry Potter for being entertaining and well-oiled.* |
| DanSumT5$_{large}$ | J. K. Rowlings Potter-filmatisering får ros af anmelderne, der roser instruktøren, men savner dramatik og en knap så nuttet hovedrolleindehaver. 
 *J. K. Rowling's Potter film adaptation is praised by critics, who praise the director but missing drama and a not-so-cute lead.* |
| DanSumT5$_{base}$ | Harry Potter-fans kan godt begynde at glæde sig, hvis de følger instruktøren David Yates' særlige præstation i 'Fantastiske skabninger og hvor de findes', mener flere anmeldere - og det er svært at se filmen, siger de. 
 *Harry Potter fans can start to rejoice if they follow the director David Yates' special performance in 'Fantastic Beasts and Where to Find Them', several reviewers believe - and it is difficult to watch the film, they say.* |
| DanSumT5$_{small}$ | J.K. Rowlings manuskript til Harry Potter-serien er særligt hendes præstation, men filmen er i fuld gang med at underholde publikum. 
 *J. K. Rowling's screenplay for the Harry Potter series is especially her achievement, but the film is busy entertaining audiences.* |

Table 8