# OpenReview forum: "DanSumT5: Automatic Abstractive Summarization for Danish"
_NoDaLiDa/2023/Conference — NoDaLiDa 2023_

### Official Review · Reviewer_zxxK · 2023-02-24
**Automatic abstractive text summarization of Danish news stories by finetuning mT5**

**Rating:** 8
**Confidence:** 4

**Review:**

This is a well written paper with experiments and thorough analysis, including limitations and negative results of approaches that did not work out. The results are decent and comparisons are made to corresponding results for other languages. The evaluation is properly done. The references list is extensive. I also very much appreciate the examples of actual data in the appendix.

A few comments:

* Since the density score has a central role (Sec. 2.1.1), maybe it could be explained in a bit more detail (not just citing the original publications)?

* In Figure 1, the two shades of green could be changed to some colors that are more easily distinguishable from each other.

* The authors state that the current paper extends previous work (Sec 1.2) by implementing "several changes to the data, model, evaluation, and fine-tuning procedure". In the final version of the paper, I would like to see a comparison to the results obtained in the previous work. Is it possible to compare the new, improved approach quantitatively and qualitatively to the previous results?


**Paper Type:**

Long paper

---

### Official Review · Reviewer_VADk · 2023-03-07
**+Much needed abstractive summarization system for Danish. -Does not compare results to any baseline.**

**Rating:** 5
**Confidence:** 4

**Review:**

## Summary
This paper describes the implementation of a Danish abstractive summarizer. It presents the results of fine-tuning three mT5-based models (based on mT5 small/base/large) on a filtered Danish summarization dataset that removes low-quality and extractive samples. The models are evaluated using automatic metrics (ROUGE and BERTScore) on a test set (~25k samples) as well as human evaluation through preference ranking between the three models and the reference on 100 random samples. Based on the results (but without directly comparing them to prior work) the paper claims that the system is **good and delivers state-of-the-art performance.** Finally, the paper discusses the model's tendency to hallucinate and make factual errors, suggesting that future efforts on Danish summarization should mirror English efforts in reducing and quantifying these phenomena.

## Review
Summarizors for the Nordic languages are long and far apart, and thoroughly documented and performant systems are highly sought after. This paper makes a great effort in this direction, providing a thorough description of the proposed system, and conducting human evaluation which is highly desired. However, the paper does not compare results to any baseline model, fundamentally questioning the results and the central claim of being "a good summarizer" and a state-of-the-art system. There is a mention of improvements over prior work, however, the paper includes no references to results or a paper to compare with. The paper possibly mentions ROUGE scores on two other languages to compare with, however, it does not make much sense to compare ROUGE across datasets (or splits), let alone different languages - this also applies to any prior work on the same dataset. Is this better than simply applying TextRank? A lead baseline? reporting these results would help determine where the proposed model stands.

Aside from this the paper is well written, exhaustively documents the system, provides interesting negative results, and an insightful analysis. In my opinion, the paper would greatly benefit from including a comparison to simple baselines (on the same data split) to asses its performance in a better-informed setting. Without this, it is unclear to me if the contribution is sufficient although there are no existing peer-reviewed summarization systems for Danish.

**Paper Type:**

Short paper

---

### Official Review · Reviewer_sMD5 · 2023-03-16
**It's alright, you get what you expect, the eval is above par**

**Rating:** 6
**Confidence:** 4

**Review:**

The paper presents a trained model on existing data, with an extra data refinement step. The model itself is novel though the procedure taken to reach its construction are not particularly novel. On the other hand, the method is not cited as a contribution of the work. Actually I'm completely missing statements of what the contribution is meant to be, or what is novel here, so am inclined to review this more as a demo paper or tech report.

ROUGE is a terrible metric so I'm glad something else is explored. It would've been good to see something like delta EI (https://aclanthology.org/2021.emnlp-main.594/) for this kind of work, above BERTscore. There is a great discussion of BERTscore which frankly is one of the highlights of the paper.

The human evaluation is great BUT who were the raters?? This evaluation is weakened without some description of that (this omission is cause for desk reject in many disciplines). Could the authors include at least the demographic information that would be present in an NLP data statement, so we can qualify the human evals predicated on the people providing those evals?

Interesting to see the comment about DanT5 performing worse. This claim is important to know, and should be quantified & substantiated in a results table.

Paper reminds me of one of my master's students, you should check out their work, "Automatic Text Summarization For Danish Using BERT"

**Paper Type:**

Demo

---

### Official Review · Reviewer_K9vL · 2023-03-17
**The paper presents a mT5-based model for Danish abstractive summarization. However, the paper seems a bit thin for a long paper and I would suggest accepting it as a short paper.**

**Rating:** 6
**Confidence:** 3

**Review:**

This paper presents the results of an effort to train abstrative summarization models for Danish based on mT5. The results of three models fine-tuned from mT5 are presented, evaluated with both automatic and human evalution procedures.

Pros:
* Advancing Danish language technology in the front of the more challenging generative task of abstractive summarization.
* In addition to automatic evaluation measures also human evaluation is used.
* The paper is mostly well written and easy to read.

Cons:
* The paper seems to have several goals: Danish abstractive summarization, problems related to evaluating summaries, discussion about general limitations of the models and data, additionally a discussion about failed experiments. Although all these topics are interesting and important, having several focuses in a single paper generally means that none of the goals are worked through with enough depth, which has also happened here. I would suggest rewriting this paper as a short paper focussing only on the abstractive summarization experiments, potentially providing more results both about the models’ variance and human evaluation and leaving the more general discussions about the other topics for the sub-sequent papers where these questions can be thoroughly investigated.
* Human evalation is only done by asking people to choose the best summary from the several summaries, which does not provide the data for understanding how high the human evaluators rated the summaries in terms of quality.

Comments:
* I don’t understand the procedure for hyperparameter tuning. Hyperparameter search was carried out by training on 2% of the data, i.e. roughly 5000 articles? The validation set was still the full set of about 25K articles? Why are the batch sizes for the hyperparameter search and for full training so different (120 vs 16)? Which hyperparameters were tuned and which values were tried? How can you be sure that the optimal hyper-parameters found on the 2% are also good on the whole data?
* Figure 1 was never referenced in the text.
* Lines 363-365 points to a reference to the authors previous work. It remains unclear what this previous work is and why is it mentioned in this context
* Table 1: Are these results based on a single run? How do we know that these differences are meaningful and not due to random noise? Adding standard deviations from several runs or CI-s from bootstrap testing would be informative.
*Lines 398-402: How are the density scores calculated?
* Section 5.1: If you want to compare with some previous results then these results should be reproduced in the current paper. Currently, with the link to the previous work reducted, it is impossible to verify that the claim, that DanSumT5-large obtains better results, is true. Reproduce these results from the previous work in Table 1.
* Section 5.2: How can I read it out from Figure 2 that DanSumT5-large summaries were preferred over the reference summaries in 21.43% cases. Maybe there should be another table giving additional results about the human evaluation.
* Based on which results did you arrive to the conclusion that DanSumT5 models generally create summaries of high subjective quality? If the human evaluation is only based on ranking then we can’t know that even the highest ranked summary was of high quality to the evaluator, we only know that it was better than the other ones.

**Paper Type:**

Long paper

---

### Decision · Program_Chairs · 2023-03-17

Accept